# An Exploratory Investigation of Organic Chemicals Detected in Baby Teeth: Differences in Children with and without Autism

## Raymond F. Palmer

Department of Family and Community Medicine, School of Medicine, University of Texas Health Science Center San Antonio, San Antonio, TX 78229, USA; palmerr@uthscsa.edu; Tel.: +1-210-827-7681

**Abstract:** Autism spectrum disorder (ASD) is a behaviorally defined neurodevelopmental disorder characterized by deficits in language, communication, and social function with an estimated prevalence rate of between 1 in 30 and 44 U.S. births. Gene/environment (G × E) interactions are widely regarded as the most probable explanation for idiopathic ASD, especially because some genes are selectively targeted by various environmental xenobiotics. Because deciduous teeth are a likely biomarker of in utero exposure, the present study investigated if the quantity of chemicals found in deciduous teeth differs between children with and without ASD. Twenty-two deciduous teeth from children with ASD and 20 teeth from typically developed children were prepared and analyzed using THE Two-Dimensional Gas Chromatography Time-of-Flight Mass Spectrometer (GC × GC-TOF MS) with ChromaTOF version 23H2 software and Agilent 7890 gas chromatograph. The autism sample had significantly more chemicals in their teeth than the typical developing sample (99.4 vs. 80.7, respectively) ($p < 0.0001$). The majority of chemicals were identified as phthalates, plasticizers, pesticides, preservatives, or intermediary solvents used in the production of fragranced personal care or cleaning products or flavoring agents in foods. The known toxic analytes reported in this study are likely biomarkers of developmental exposure. Why there were greater concentrations of toxic chemicals in the teeth that came from children with ASD is unclear. A further understanding of the cavalcade of multiple biological system interactions (Interactome) could help with future efforts to reduce risks. Notwithstanding, the avoidance of pesticides, plastics, and scented personal care products may be warranted under the precautionary principle rule.

**Keywords:** environmental toxic exposure; pesticides; deciduous teeth; autism





## 1. Introduction

Autism spectrum disorder (ASD) is a behaviorally defined neurodevelopmental disorder characterized by deficits in language, communication, and social function [1]. The most recent prevalence estimates range from 1 in 30 to 44 U.S. births [1–3].

Currently, gene/environment (G × E) interactions are widely regarded as the most probable explanations for idiopathic ASD [4–6]. With genes that are selectively targeted by various environmental xenobiotics [7], the interface between the emerging genome and exposome sciences complicates the understanding of interactions between an individual's predisposed biology and multiple environmental exposures [6–8].

The U.S. Toxic Substances Control Act (TSCA, 1976) considered approximately 62,000 chemicals that were not subject to testing or regulation unless proven to "present(s) an unreasonable risk of injury to health or the environment [9,10]". Now, the number of untested chemicals is over 83,000 and growing [10]. Environmental health monitoring studies reveal multiple scientific challenges in predicting and preventing disease. As such, there is a widely held consensus for the reform of outdated TSCA laws [11–14].

The National Health and Nutrition Examination Survey (NHANES) confirms that the exposure of the U.S. population to toxic environmental chemicals has increased over

the last 40 years [15] and that exposures to ubiquitous neurotoxins affect women of child-bearing age and their infants [16–21]. The developing fetus is particularly vulnerable to adverse effects from toxic environmental chemical exposures, including polycyclic aromatic hydrocarbons, phthalates, pesticides, organophosphates, pyrethroids diester phthalates, and polybrominated diphenyl ethers—there is a broad range of studies in the literature and a clear link between in utero chemical exposures, immune system dysregulation, endocrine disruption, and impaired childhood neurodevelopment, including autism [22–32].

While the aforementioned studies measured current exposure levels, no biomarkers have existed to retrospectively assess early exposure to many organic chemicals, including organophosphate pesticides and diester phthalates, since the parent compound is rapidly metabolized, and the metabolites are quickly excreted.

Measuring Exposure: blood, urine, saliva, or hair has often been used to assess the risk of disease by comparing the concentrations of some environmental toxic substances between affected and unaffected individuals [33,34] While this has been a useful approach, these bio-samples are only measures of recent exposure and cannot inform about distant past or developmental exposures, particularly during critical neurodevelopmental periods. Alternatively, deciduous teeth provide a historical record of exposure with the potential to circumvent the limitation of biomarkers commonly used in epidemiological studies, such as blood and urine, which only capture current exposures in a child [35].

The use of deciduous tooth crowns has served as a biomarker of early developmental exposure [35,36], including a biomarker that showed atypical fetal inflammatory regulation among those with ASD [37]. The mineralization of primary teeth begins prenatally between 14 and 16 weeks' gestation and concludes postnatally from 1.5 to 3 months for incisors, 9 months for canines, and 5.5 to 11 months for molars [38]. Exogenous and endogenous organic chemicals or their metabolites circulating in the bloodstream absorb into the developing tooth and remain stored thereafter. The tooth concentrations of these analytes are likely biomarkers of exposure during the period of tooth crown formation, although a fraction likely reflects later post-natal exposure [35–37].

It has been demonstrated that metals in circulation, which are present during the period of tooth formation, become incorporated into forming dental tissue and are stored in the mineral component of teeth [39]. Several studies demonstrate the advantage of primary tooth crown analysis for determining exposures that might be related to various disease outcomes. Indeed, the conclusions from a 2010 NATO workshop on the effects of heavy metal pollution on child development recommend that depositions found in teeth can serve as a valuable tool in relating heavy metal pollution to childhood development outcomes [40].

The vast majority of studies using deciduous teeth have been on measuring heavy metal concentrations. Less focus has been placed on the assessment of organic toxicant exposures such as pesticides, plastics, or pharmaceuticals. Our laboratory has previously discovered, with replication in two cohorts, that analgesics (acetaminophen, ibuprofen), which are specific metabolites of organophosphate pesticides and phthalates (chlorpyrifos, diazinon, dibutyl phthalate, and DEHP), and the insect repellant DEET, remain stored in pulverized deciduous molars in typically developing children at levels detectable by Liquid Chromatography Tandem Mass Spectrometry (LC/MS/MS) [41,42]. Prior to this, semi-volatile organic chemicals (SVOCs) had not been identified in teeth. While methods with sectioned teeth have recently been perfected for the timing of SVOC detection [35,36,43], in this study, the precise timing of exposure was not determined. This study used GC × GC-TOF MS methods with pulverized teeth to identify new chemicals and to quantify the amounts between children with and without ASD.

The present exploratory study investigated an untargeted approach to identify new SVOC chemicals and determine if chemical exposure patterns differed between children with ASD and typically developing children. Our preliminary hypothesis was that more harmful chemicals would be present in the teeth from children with autism compared to those without autism.

## 2. Materials and Methods

### 2.1. Sample and Recruitment

We established a tooth repository of over 900 teeth with survey information on demographics, environmental exposures, and developmental medical histories. Our tooth biorepository (from 280 mothers with 363 total children) mostly consisted of teeth donated by the families of children with ASD currently residing throughout the US who participated in the Interactive Autism Network (IAN). The IAN network is the nation's largest online autism research forum, where over 43,000 families complete comprehensive surveys and participate in various research studies [44,45].

The inclusion criteria for IAN participants are parents with children under 18 years of age who have been diagnosed with ASD by a health professional. The diagnosis of ASD in the IAN database has been clinically validated in a subsample of participants [44] as well as verified by a review of parent- and professional-provided medical records [45]. This work reports the 98% concordance of validated ASD diagnoses with parental self-report, thus supporting the viability of the centralized database recruitment model. In a recently completed project to verify diagnosis in our non-IAN recruits, a certified diagnostician used a subset of our sample and confirmed that either parent provided (1) medical records and/or diagnostic workups that had been previously completed by a qualified health professional, or (2) an ADIR interview was conducted by phone if verifiable records could not be obtained.

In the current study, 42 teeth were randomly selected from our repository (22 from children with ASD and 20 from typically developing children). The careful selection of age–gender-matched controls for each case was used. Each participant donated between 1 and 10 teeth. Every attempt was made to match tooth types (molars, incisors, canines) between cases and controls. We excluded the use of teeth of children whose mothers reported having children with various other medical or neurological conditions, including ADHD, and retained only those with ASD and those that were pure controls (e.g., typically developing children with no reported medical, emotional, or psychological comorbidities). All sample collection was approved by the University of Texas Health Science Internal Review Board (#HSC 20110313H, approved 13 May 2011).

### 2.2. Laboratory Methods

Teeth were prepared and analyzed as described by Camann et al. [40] at the Southwest Research Institute (SWI) in San Antonio, TX, USA. Any attached roots were severed from the tooth crown at the cemento-enamel junction with a heatless wheel. The pulp was removed from the pulp chamber with a drill bit. The crown was gently swirled in dichloromethane (DCM), and the wash was retained as a quality measure to evaluate external tooth contamination. Each tooth crown was pulverized to a fine powder with a clean mortar and pestle, and the powder was weighed.

A 25 mg aliquot of each tooth powder sample was spiked with 1-methylnaphthalene-d10, p-terphenyl-d14, and di-n-pentyl phthalate-d4 as extraction surrogates and was extracted in 1 mL of DCM for 30 min in a water bath sonicator, before being cooled with ice, and the extract was concentrated to 250 µL. A matrix blank consisting of 25 mg of pulverized kiln-fired synthetic hydroxyapatite and a solvent blank were prepared and extracted along with each extraction batch of powdered tooth aliquots.

Prior to analysis, tooth extracts and the matrix and solvent blanks were spiked with isotopically labeled internal standards, including 1,4-dichlorobenzene-d8, acenaphthene-d10, anthracene-d10, chrysene-d12, and perylene-d12. Analysis was performed on a LECO Pegasus 4D Two-Dimensional Gas Chromatography Time-of-Flight Mass Spectrometer (GC × GC-TOF MS) with Chromatof software and the Agilent 7890 gas chromatograph. The instrument was fitted with two chromatographic columns of distinct phases connected in series.

GC × GC-TOF MS operates similarly to a one-dimensional GC-MS, except there are two analytical gas chromatography columns connected in series. As the compounds elute

from the first column, they pass through a modulator which acts to trap and release the compounds onto the dissimilar, shorter second column. A Time-of-Flight analyzer is used to provide the required high speed needed to define the chromatographic peaks, which are rendered narrower by the modulator. The first column, Restek Rxi-1MS (250 μm × 0.25 μm, 30 m), separates the compounds based on their boiling point, while the second column, Restek Rxi-17SilMS (Centre County, PA, USA) (180 μm × 0.18 μm, 1.3 m), performs an additional separation based roughly on polarity and double bond equivalents. This orthogonal technique results in the high-resolution chromatographic separation of compounds, resulting in fewer interferences and higher-quality mass spectra over a traditional one-dimensional GC/MS.

For preliminary screening, the National Institute of Standards and Technology (NIST) mass spectral library was used for peak identification. The detections of potential interest based on library searching were compared to the matrix and solvent blanks; retained detections are present at a level at least 10× greater than the solvent and matrix blanks. The quality of the NIST identifications was reviewed by mass spectral experts to create a list of candidate compounds for retention time confirmation. Up to 40 analytical standards for the candidate compounds were procured through our existing inventory or purchased from commercial vendors. The retention times and mass spectra of the analytical standards were compared to those measured in the tooth extracts. Candidate compounds matching the corresponding standard in both the retention time and mass spectrum were qualitatively confirmed and subsequently quantified in the teeth using linear regression. High-Performance Liquid Chromatography Quadrupole Time-of-Flight (HPLC/qTOF) Mass Spectrometry was also used in the event that additional confirmation of a compound's accurate mass is required. The relevance of the aforementioned untargeted analytic approach and more specific details of the methods used in this paper have been published elsewhere by the SWRI lab [46,47].

The product use categories of the identified chemicals were classified using PubChem [48] and other databases, including the Good Sense Company (https://www.thegoodscentscompany.com) (accessed on 11 March 2024), U.S. Environmental Protection Agency Chemical Assessment Summary-Integrated Risk Information System (IRIS), and National Center for Environmental Assessment (https://iris.epa.gov) (accessed on 3 December 2024).

### 2.3. Statistical Analysis

Comparisons of the chemical concentration between cases and controls were made using *t*-tests with $p < 0.05$ as the critical cutoff for significance. The t-distribution is most useful for small sample sizes, when the population standard deviation is not known, or both.

### 2.4. Statistical Power

Sample sizes of 20–30 with a minimum of 12 for pilot studies have been recommended by various research statisticians [49,50]. In this pilot study ($n = 22$ ASD group, and $n = 20$ for the reference group), the power to detect a between-group medium effect size was relatively low (power = 0.71, using two-tail, alpha = 0.05). However, there was 80% power to detect a moderately large effect size (0.6).

Pilot/exploratory studies are performed prior to definitive trials to provide enough evidence of their overall potential intervention benefits [51,52]. Schoenfeld [53] suggests that preliminary hypothesis testing for efficacy could be conducted with a high type I error rate (a false positive rate up to $p < 0.25$). Notwithstanding, in this study, we still accept a conservative alpha of $p < 0.05$. Pursuant to avoid missing a true effect (Type II error), we did not adjust for multiple comparisons. All analyses were performed with SAS software V.9.4 [54].

## 3. Results

Table 1 shows the sample demographics. Due to initial matching, there were no significant differences between the cases and controls for mothers' age, child's gender or ethnicity, and household income. Therefore, covariate adjustment in the statistical analysis was unnecessary.

**Table 1.** Sample demographics.

|  | Cases (*n* = 22) | Controls (*n* = 20) | *p* |
|---|---|---|---|
|  | Mean (SD) or % | Mean (SD) or % |  |
| Mothers Age | 40.4 (6.4) | 41.1 (5.3) | 0.77 |
| Child Gender (% male) | 80.1% | 70.1% | 0.67 |
| Family Income |  |  |  |
| Less than USD 15,000 | 6.7% | 18.5% |  |
| USD 15,000–USD 24,999 | 86.7% | 44.4% |  |
| USD 25,000–USD 34,999 | 6.7% | 11.1% |  |
| Not reported | 0% | 25.9% | 0.61 |
| Ethnicity |  |  |  |
| Hispanic | 6.7% | 22.2% | 0.09 |
| Non-Hispanic White | 60.0% | 44.4% |  |
| Other | 26.7% | 7.4% |  |
| Not Reported | 6.6% | 25.9% |  |

A total of 11,971 chemicals were found, including unknowns without a suitable library match. The average was 315 compounds per tooth. A total of 201 compounds of interest were identified and confirmed as positive matches in Chemical Abstract Services (CASs) libraries. In general, these were exogenous xenobiotic compounds with a high identity confidence score. The first row in Table 2 shows that the autism sample had significantly more chemicals in these teeth than the teeth from the typically developing sample (99.4 vs. 80.7, respectively) ($p < 0.0001$). The distribution is shown graphically in Figure 1.

Of the 201 identified compounds, one-fourth of these chemicals (*n* = 54 or 27%) were statistically distinguishable between teeth from those with and without autism. Table 3 shows the means, standard deviation, and minimum/maximum range of each of these chemicals. The overwhelming majority of these chemical concentrations were higher in the teeth from the autism sample. There were only four chemicals where the teeth of typically developing children were higher. These were Limonene (a citrus flavoring or fragrant agent), Benzothiazole (antimicrobial), Butyl benzoate (antimicrobial), and Butylated Hydroxytoluene (preservative/food additive and antioxidant).

**Table 2.** Total number of chemicals and total micrograms/grams (µg/g) between children with and without ASD.

|  | Autism | | Typically Developing Controls | | *p* |
|---|---|---|---|---|---|
|  | Mean (SD) | Min–Max | Mean (SD) | Min–Max |  |
| Total number of chemicals | 99.41 (12.78) | 75–129 | 80.65 (13.18) | 62–115 | 0.0001 |
| Total grams | 4799.44 (2570.36) | 1388.16–11,332.95 | 3192.90 (1603.39) | 606.25–7707.95 | 0.01 |

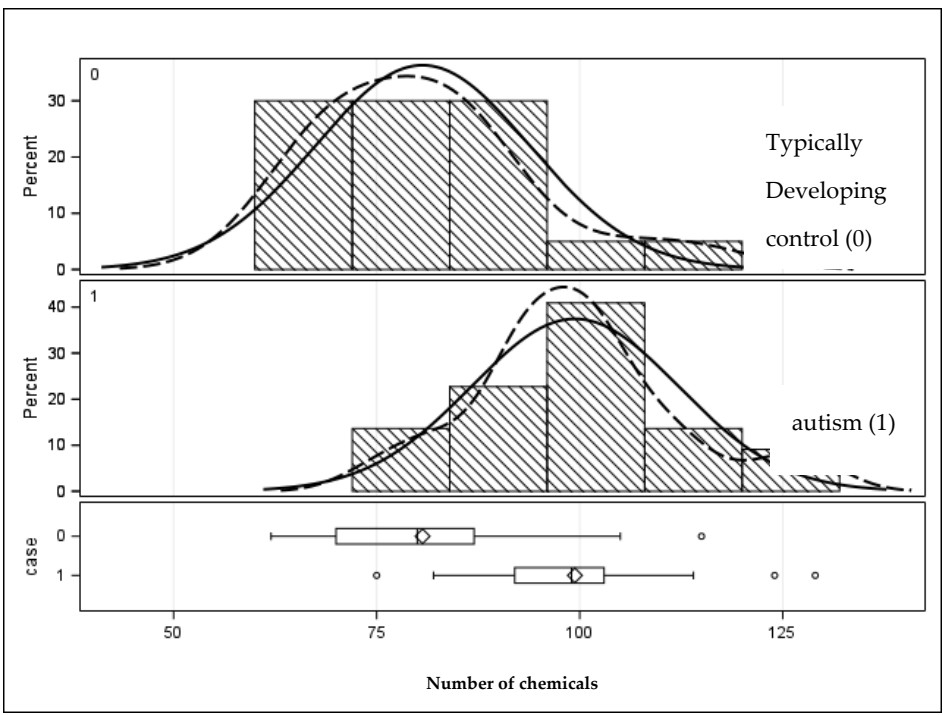

**Figure 1.** Distribution of the number of chemicals in the primary teeth of children with and without autism. Solid lines represent the population distribution, and the dashed lines indicate the sample distribution. Given they are a close match signifies that the sample is sufficiently well represents the population.

**Table 3.** Chemicals (μg/g) that statistically differ between Autism cases and controls (55 out of 202).

| Chemical | Autism Cases | | | | Typically Developing Child Cases | | | |
|---|---|---|---|---|---|---|---|---|
| | Mean | (Std Dev) | Minimum | Maximum | Mean | (Std Dev) | Minimum | Maximum |
| Chem1 | 0.10 | (0.17) | 0.00 | 0.49 | 0.01 | (0.03) | 0.00 | 0.15 |
| Chem2 | 0.32 | (0.26) | 0.00 | 0.99 | 0.11 | (0.18) | 0.00 | 0.77 |
| Chem3 | 0.51 | (0.60) | 0.00 | 1.84 | 0.16 | (0.32) | 0.00 | 1.05 |
| Chem4 | 1.97 | (1.69) | 0.00 | 7.43 | 0.77 | (1.31) | 0.00 | 5.62 |
| Chem5 | 0.22 | (0.32) | 0.00 | 0.99 | 0.07 | (0.15) | 0.00 | 0.43 |
| Chem6 | 1.08 | (1.23) | 0.00 | 5.13 | 0.30 | (0.52) | 0.00 | 2.00 |
| Chem7 | 0.23 | (0.27) | 0.00 | 0.80 | 0.03 | (0.09) | 0.00 | 0.33 |
| Chem8 | 1.49 | (1.67) | 0.17 | 7.72 | 0.25 | (0.42) | 0.00 | 1.47 |
| Chem9 | 0.15 | (0.24) | 0.00 | 0.78 | 0.04 | (0.16) | 0.00 | 0.68 |
| Chem10 | 15.23 | (22.83) | 0.00 | 49.71 | 2.36 | (10.56) | 0.00 | 47.24 |
| Chem11 | 6.75 | (6.12) | 0.00 | 19.38 | 2.30 | (3.36) | 0.00 | 9.76 |
| Chem12 | 87.56 | (109.22) | 0.00 | 538.23 | 27.17 | (27.88) | 0.00 | 116.99 |
| Chem13 | 1.04 | (0.87) | 0.29 | 3.41 | 0.56 | (0.49) | 0.00 | 2.20 |
| Chem14 | 0.40 | (0.53) | 0.00 | 2.36 | 0.16 | (0.37) | 0.00 | 1.44 |
| Chem15 | 3.38 | (2.88) | 0.94 | 11.76 | 5.29 | (3.18) | 0.00 | 13.88 |
| Chem16 | 0.24 | (0.49) | 0.00 | 1.87 | 1.41 | (1.76) | 0.00 | 6.18 |
| Chem17 | 3.08 | (2.12) | 0.98 | 10.03 | 2.01 | (1.63) | 0.00 | 6.60 |
| Chem18 | 45.60 | (29.33) | 0.00 | 111.20 | 12.20 | (15.62) | 0.00 | 61.78 |

**Table 3.** *Cont.*

| Chemical | Autism Cases | | | | Typically Developing Child Cases | | | |
| --- | --- | --- | --- | --- | --- | --- | --- | --- |
| | Mean | (Std Dev) | Minimum | Maximum | Mean | (Std Dev) | Minimum | Maximum |
| Chem19 | 1.37 | (2.45) | 0.00 | 10.89 | 0.21 | (0.33) | 0.00 | 1.00 |
| Chem20 | 1.40 | (1.31) | 0.19 | 5.83 | 0.59 | (0.53) | 0.00 | 1.65 |
| Chem21 | 3.32 | (3.31) | 0.58 | 15.02 | 1.86 | (2.03) | 0.00 | 8.58 |
| Chem22 | 0.03 | (0.09) | 0.00 | 0.31 | 0.39 | (0.77) | 0.00 | 2.77 |
| Chem23 | 1.17 | (0.60) | 0.33 | 2.90 | 3.45 | (5.01) | 0.41 | 19.98 |
| Chem24 | 0.90 | (0.97) | 0.00 | 3.36 | 0.25 | (0.95) | 0.00 | 4.21 |
| Chem25 | 1.22 | (0.62) | 0.60 | 2.97 | 0.83 | (0.51) | 0.00 | 2.11 |
| Chem26 | 4.03 | (5.63) | 0.00 | 18.09 | 0.85 | (1.57) | 0.00 | 5.98 |
| Chem27 | 1.19 | (0.75) | 0.00 | 2.80 | 0.15 | (0.37) | 0.00 | 1.40 |
| Chem28 | 0.24 | (0.29) | 0.00 | 1.18 | 0.06 | (0.15) | 0.00 | 0.61 |
| Chem29 | 0.10 | (0.26) | 0.00 | 1.09 | 0.00 | (0.00) | 0.00 | 0.00 |
| Chem30 | 0.37 | (0.45) | 0.00 | 1.87 | 0.08 | (0.16) | 0.00 | 0.62 |
| Chem31 | 8.26 | (11.79) | 0.00 | 30.80 | 0.26 | (0.84) | 0.00 | 3.27 |
| Chem32 | 0.20 | (0.22) | 0.00 | 0.76 | 0.07 | (0.16) | 0.00 | 0.69 |
| Chem33 | 7.90 | (6.86) | 0.00 | 30.65 | 1.53 | (2.04) | 0.00 | 6.28 |
| Chem34 | 0.40 | (0.45) | 0.00 | 1.61 | 0.07 | (0.17) | 0.00 | 0.60 |
| Chem35 | 13.55 | (14.14) | 0.43 | 55.87 | 3.81 | (9.86) | 0.00 | 44.85 |
| Chem36 | 0.26 | (0.24) | 0.00 | 1.04 | 0.14 | (0.19) | 0.00 | 0.59 |
| Chem37 | 104.19 | (84.68) | 40.70 | 364.51 | 34.96 | (39.04) | 0.00 | 149.21 |
| Chem38 | 5.08 | (12.02) | 0.00 | 44.51 | 23.79 | (27.90) | 0.00 | 119.15 |
| Chem39 | 7.55 | (8.19) | 1.94 | 40.46 | 3.77 | (2.19) | 0.46 | 9.13 |
| Chem40 | 3.65 | (3.16) | 0.00 | 12.24 | 1.63 | (1.92) | 0.00 | 7.02 |
| Chem41 | 3.49 | (2.51) | 0.00 | 11.00 | 0.92 | (1.94) | 0.00 | 8.27 |
| Chem42 | 2.96 | (7.63) | 0.00 | 22.41 | 0.00 | (0.00) | 0.00 | 0.00 |
| Chem43 | 6.90 | (11.73) | 0.67 | 57.12 | 2.16 | (2.55) | 0.00 | 11.11 |
| Chem44 | 139.17 | (103.90) | 0.00 | 339.60 | 75.98 | (117.41) | 0.00 | 394.04 |
| Chem45 | 5.73 | (7.64) | 0.86 | 35.25 | 1.46 | (1.64) | 0.00 | 6.26 |
| Chem46 | 340.09 | (485.55) | 30.49 | 1966.46 | 59.84 | (58.48) | 4.53 | 244.78 |
| Chem47 | 35.42 | (28.77) | 5.63 | 122.78 | 17.25 | (29.99) | 0.48 | 136.16 |
| Chem48 | 3.16 | (3.20) | 0.38 | 12.29 | 1.43 | (2.03) | 0.00 | 8.99 |
| Chem49 | 3756.79 | (2500.01) | 895.83 | 10,276.18 | 2643.87 | (1612.12) | 376.14 | 6898.29 |
| Chem50 | 2.40 | (2.86) | 0.00 | 10.10 | 0.34 | (0.67) | 0.00 | 1.99 |
| Chem51 | 6.31 | (9.62) | 0.32 | 44.91 | 0.99 | (2.23) | 0.00 | 9.99 |
| Chem52 | 1.64 | (4.51) | 0.00 | 17.88 | 0.00 | (0.00) | 0.00 | 0.00 |
| Chem53 | 1.59 | (1.92) | 0.00 | 9.14 | 0.41 | (0.65) | 0.00 | 1.90 |
| Chem54 | 75.20 | (51.88) | 0.00 | 189.40 | 32.07 | (39.70) | 0.00 | 126.47 |

The bottom row of Table 2 summarizes the results of Table 3, showing that teeth from children with autism had statistically greater concentrations (µg/g) than children without autism ($p < 0.01$). Figure 2 shows the graphic distribution. It can also be seen that the upper

range (Max) of the number and concentration of chemicals is higher in the teeth from the autism sample.

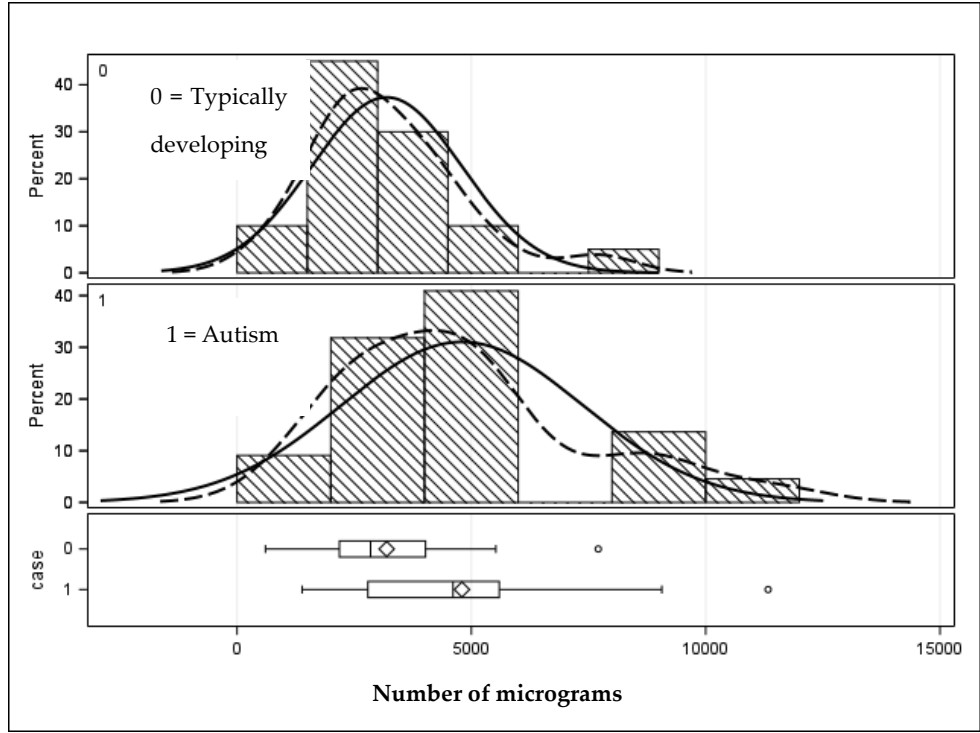

**Figure 2.** Distribution of micrograms (μg/g) of chemicals distinguishing children with and without autism. Solid lines represent the population distribution, and the dashed lines indicate the sample distribution. Given they are a close match signifies that the sample is sufficiently well represents the population.

The name and product usage of the 54 chemicals that were significantly different between the 2 groups (reflected in Table 3) can be seen in Supplementary Table S1. The inspection of this table shows that the majority of chemicals are classified as phthalates, plasticizers, pesticides/microbiocides, antimicrobials, preservatives, or intermediary solvents used in the production of these compounds. Five of the chemicals were identified by the Pesticide Action Network (PAN) as *PAN Bad Actors*. These pesticides are at least a known or probable carcinogen, reproductive or developmental toxicants, a neurotoxic cholinesterase inhibitor, a known groundwater contaminant, or a pesticide with high acute toxicity. Many of the chemicals found in this study are primarily used as fragrance compounds for personal care, cleaning products, or flavoring agents in foods.

## 4. Discussion

Using an exploratory untargeted mass spectrometry approach, a broad array of chemicals in children's deciduous teeth were captured. Since the mineralization of primary teeth begins prenatally [38], both exogenous and endogenous organic chemicals or their metabolites circulating in the maternal bloodstream absorb into the developing tooth and remain stored thereafter. The tooth concentrations of the analytes reported in this study are likely biomarkers of developmental exposure during the period of tooth crown formation, although a fraction may also reflect later post-natal exposure [35,36].

From the over 11,000 chemicals that were captured, there were 201 exogenous chemicals that were positive matches identified in the known chemical abstracts with Chemical Abstract Services (CASs) as unique identifiers. One-fourth of those 201 chemicals (55 or 27%) were statistically distinguishable between teeth from those with and without autism. Of those 55 teeth, all but 4 were significantly elevated in the teeth of children

with ASD. Why there were more chemicals found in the teeth of ASD children remains unclear. One plausible explanation may be due to group differences in the ability to metabolize xenobiotics [55,56].

Overall, the teeth of children with autism had significantly more and greater concentrations of chemicals—primarily from pesticides and plastics/polymers often found in food, cosmetics, or household products. This aligns well with the literature implicating that these compounds increase the risk of autism. Pesticides and plastics are neurotoxins and endocrine disruptors, from which systematic reviews suggest that gestational exposure to pesticides is linked to autism risk [57–59]. Similarly, phthalate plasticizers have been shown to affect neurodevelopment with adverse consequences, including autism and ADHD risk [60,61].

The effects of xenobiotic exposure and its role in autism are rapidly evolving, and gene/environment interactions are now considered a rich area of research [62–64]. This is particularly relevant given the report by Carter and Blizard [7], who report that autism genes are selectively targeted by environmental pollutants, including pesticides and phthalates found in food, cosmetics, or household products.

This study is consistent with previous reports demonstrating that pesticides and plastics are found in the teeth of children with autism [41,42]. While the exploratory results of this study are preliminary and lack genetic information, our primary hypothesis was supported and is aligned with the literature showing an increased risk of autism from exposure to neurotoxicants and endocrine disruptors during development.

The genome has opened a pandora's box of possibilities [65,66] in terms of interacting environmental and biological risk factors that act to increase the risk of autism. The resulting impact on gene expression and interactions between the Proteome, Metabolome, and Microbiome [65,67] adds massive complexity. Furthermore, the interactive effect of the exposome on these systems can be acknowledged, all coming together as a vast Interactome [68,69]. While it will take time to fully understand these processes and how they interact to mount prevention efforts and reduce the risk of ASD, it may not be necessary to understand all the biological components to immediately mount risk reduction strategies.

The precautionary principle is a conservative decision rule used to reduce the risk of disease when the risk is not completely known but has the potential for critical damage [70]. While some debate exists due to the principle's vague definition [71], in many instances, it seems relevant to take preventative action to prevent harm, even if the likelihood of harm is uncertain [68]. A relevant example of autism includes the avoidance of artificial sweeteners during pregnancy. Fowler et al. [72] have shown that aspartame in diet sodas and other artificial chemical sweeteners is associated with a threefold risk for ASD and ADHD. A mother's chemical intolerance status may even further compound the risk of ASD [73].

Conclusions: In light of the findings from this study, the avoidance of pesticides, plastics, and scented personal care products [74] may be warranted under the precautionary principle rule. This is highly relevant for women of childbearing age who want to have children. There is a need to share information with the lay public and among healthcare professionals on how best to avoid toxic exposures in order to reduce the risk of neurological impairment during development.

**Supplementary Materials:** The following supporting information can be downloaded at: https://www.mdpi.com/article/10.3390/jox14010025/s1, Table S1: Chemical function and product usage.

**Funding:** This study was funded in part by Autism Speaks Grant no. 8426, the Suzanne and Bob Wright Trail Blazer award, granted to the first author.

**Institutional Review Board Statement:** The study was conducted in accordance with the Declaration of Helsinki and approved by the Institutional Review Board of the University of Texas Health Science Internal Review board (#HSC 20110313HU) (approved 13 May 2011).

**Informed Consent Statement:** Written informed consent was obtained from the participants in this study as required by the Institutional Review Board of the University of Texas Health Science Internal Review board (#HSC 20110313HU) (approved 13 May 2011).

**Data Availability Statement:** Data are available by reasonable request to the author.

**Conflicts of Interest:** The author declares no conflict of interest. The funders had no role in the design of the study; in the collection, analyses, or interpretation of data; in the writing of the manuscript; or in the decision to publish the results.

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
