# Peer review of "An Exploratory Investigation of Organic Chemicals Detected in Baby Teeth: Differences in Children with and without Autism"

_jox, doi:10.3390/jox14010025_

Round 1

Reviewer 1 Report

Comments and Suggestions for Authors

Introduction says:We used LC/MS/MS methods with pul- 91 verized teeth to identify new chemicals and to quantify the amounts between children 92 with and without ASD. However experimental says that  GC/MS was used .

The number of the specific theet used is not specified, but as they have different sizes, this parameter should be considered, and the same theet should be compared (molar or inciseve or...)

Table 2, figures are with and without commas.

Table 3 how are concentrations expressed? the are no units mentioned 

Conclusions should be added

Comments on the Quality of English Language

good

Author Response

Thank you for your helpful comments.

Reviewer 1

Introduction says: We used LC/MS/MS methods with pul- 91 verized teeth to identify new chemicals and to quantify the amounts between children 92 with and without ASD. However experimental says that  GC/MS was used .

This has been correct (line 93)

The number of the specific theet used is not specified, but as they have different sizes, this parameter should be considered, and the same theet should be compared (molar or inciseve or...)

We have added how this was addressed in the methods section (line 123)

Table 2, figures are with and without commas.

              Commas are now used if the number had 5 digits

Table 3 how are concentrations expressed? the are no units mentioned 

              These units are now included on the tables 2 and 3, figure 2, and on line 240

Conclusions should be added

              This has been added starting on line 311.

Reviewer 2 Report

Comments and Suggestions for Authors

Because deciduous teeth are a likely biomarker of in utero exposure, authors investigate if the quantity of chemicals found in deciduous teeth differ between children with and without autism spectrum disorder (ASD). Results showed greater concentrations of toxic chemicals in the teeth that came from children with ASD, although the reason is still unclear.

I found the manuscript to be well written and the results to be very interesting. 

 I only have some minor comments: 

In page 1 line 9 authors stated, “1 in 36 births” but in line 31 wrote “1 in 30 - 44 U.S. births (1-3).” Please unify. 

Did authors consider isolating dentin layers from desired developmental time periods? Some authors have found significant differences between these sections.  For example, In Arora and Austin, 2013 (Teeth as a biomarker of past chemical exposure), it was determined that there was a significant positive association of Mn levels in parts of dentine formed in the second trimester with Mn loading in floor dust sampled during the second trimester of pregnancy. 

Considering that authors refer to have collected data on environmental exposures, these collected data with exposure levels from distinct dentine layers could elucidate more about differences between the two studied groups.

In figure 1. It is missing the label ASD and Typically developing.

I would suggest that table 4 would be sent to supplementary material.

In discussion authors should refer to “survey information on demographics, environmental exposures, and developmental medical histories” (stated in page 3 lines 100-101). I essentially see discussion based on demographics. Which environmental exposures information authors have collecte? and why not assess if those collect data make a difference to the exposure levels.

Comments on the Quality of English Language

In section materials and methods authors often used the infinitive tense making it difficult to understand if it is something that was performed in this study or will be performed in future studies.

Examples: 

“Analysis will be performed on a LECO Pegasus 4D Two-Dimensional Gas…”. 

“Up to 40 analytical standards for the candidate compounds will be procured through our existing inventory or purchased from commercial vendors. The retention times and mass spectra of the analytical standards will be compared to those measured in the tooth extracts. …”

My question is Will be or was performed in the current study. Please clarify and rephrase it.

Author Response

Reviewer 2: Thank you for your most helpful comments and insight.

Because deciduous teeth are a likely biomarker of in utero exposure, authors investigate if the quantity of chemicals found in deciduous teeth differ between children with and without autism spectrum disorder (ASD). Results showed greater concentrations of toxic chemicals in the teeth that came from children with ASD, although the reason is still unclear.

I found the manuscript to be well written and the results to be very interesting. 

 I only have some minor comments: 

In page 1 line 9 authors stated, “1 in 36 births” but in line 31 wrote “1 in 30 - 44 U.S. births (1-3).” Please unify. 

              This has been made consistent (line 9 and 31)

Did authors consider isolating dentin layers from desired developmental time periods? Some authors have found significant differences between these sections.  For example, In Arora and Austin, 2013 (Teeth as a biomarker of past chemical exposure), it was determined that there was a significant positive association of Mn levels in parts of dentine formed in the second trimester with Mn loading in floor dust sampled during the second trimester of pregnancy. 

Considering that authors refer to have collected data on environmental exposures, these collected data with exposure levels from distinct dentine layers could elucidate more about differences between the two studied groups.

              An excellent observation, in this study, we used pulverized tooth crowns to assess total amounts of chemicals. To identify the timing of targeted analytes using laser ablation at different dentine layers requires a proprietary methodology used in Dr Aroras work. This involves fine slices of teeth which are then subject to laser ablation. Therefore, pulverized teeth cannot be used determine timing of exposure. Further, laser ablation methods cannot be used to identify volatile organic compounds, as the VOCs do not withstand the laser ablation. Aroras method requires that specific analytes be targeted and exploratory investigations, such as our focus here, cannot be performed.

I have co-authored a few papers with Dr Arora to investigate the timing of exposure to different metals. The purpose of the current paper was to identify as many compounds as possible, and as mentioned above, identifying timing is not possible at this time.

In figure 1. It is missing the label ASD and Typically developing.

              This has now been added.

I would suggest that table 4 would be sent to supplementary material.

This is a good idea. I have moved this to the supplementary material. Language has been revised at line 249 to reflect this change.

In discussion authors should refer to “survey information on demographics, environmental exposures, and developmental medical histories” (stated in page 3 lines 100-101). I essentially see discussion based on demographics. Which environmental exposures information authors have collecte? and why not assess if those collect data make a difference to the exposure levels.

A great point. However, only select targeted self-reported environmental exposures were assessed. Given the exploratory nature of this paper, the self-reported information would not add to the message of the present manuscript, as many different compounds were identified. In a previous published paper, we targeted specific chemicals to verify if they indeed corresponded to the small set of self-reported exposures. (if interested, please see Palmer et al., Organic Compounds Detected in Deciduous Teeth: A Replication Study from Children with Autism in Two Samples. J Environ Public Health. 2015; 2015:862414. https://www.hindawi.com/journals/jeph/2015/862414/

Comments on the Quality of English Language

In section materials and methods authors often used the infinitive tense making it difficult to understand if it is something that was performed in this study or will be performed in future studies.

Thank you. This has all been corrected to the present tense.